# Genetic Diversity of Porcine Group A Rotavirus Strains from Pigs in South Korea

**DOI:** 10.3390/v14112522

**Published:** 2022-11-14

**Authors:** Gyu-Nam Park, Da In Kim, SeEun Choe, Jihye Shin, Byung-Hyun An, Ki-Sun Kim, Bang-Hun Hyun, Jong-Soo Lee, Dong-Jun An

**Affiliations:** 1Virus Disease Division, Animal and Plant Quarantine Agency, Gimcheon 39660, Gyeongbuk-do, Republic of Korea; 2College of Veterinary Medicine, Chungnam National University, 220 Gung-dong, Daejeon 34134, Yuseong-gu, Republic of Korea; 3Department of Veterinary Medicine Virology Laboratory, College of Veterinary Medicine and Research Institute for Veterinary Science, Seoul National University, Gwanak-ro 1, Seoul 08826, Gwanak-gu, Republic of Korea

**Keywords:** porcine rotavirus, genetic diversity, prevalence, genotypes, isolation

## Abstract

Porcine group A rotavirus (PoRVA; family, *Reovirideae*) strains cause acute viral gastroenteritis in piglets (especially suckling and weaned pigs), resulting in significant economic losses. In this study, we analyzed the VP7 and VP4 genes of PoRVA isolated between 2014 and 2018 from domestic pigs in South Korea to investigate the prevalence of predominant circulating genotypes (G and P types). The prevalence of the PoRVA antigen in the diarrheic fecal samples was 14.1% (53/377). Further genetic characterization of the VP7 and VP4 genes of 53 PoRVA isolates identified six different G-genotypes and five different P genotypes. The G4 and G9 genotypes were the most common (each 39.6%) in PoRVA-positive pigs, followed by P[7] and P[6] (33.9% and 30.1%, respectively). Because the G5 and G9 genotype vaccines are currently mainly used in South Korea, this result provides valuable epidemiological information about the genetic characteristics of PoRVA circulating on domestic pig farms. Development of a novel PoRVA vaccine that targets the current strains circulating in South Korea may be required for more effective virus control on pig farms.

## 1. Introduction

Severe and acute dehydrating diarrhea is a major cause of high mortality and morbidity in young animals, including piglets. In the swine industry, this enteric disease causes serious economic losses through deaths, decreased growth rates, and increased treatment costs [1,2,3,4]. Multiple viruses can cause acute infectious diarrhea [1,2,5]. Rotaviruses (RVs), which belong to the family *Reoviridae* (genus *Rotavirus*), are important pathogens that cause viral gastroenteritis in piglets worldwide [2,6]. Based on the antigenic properties and sequence diversity of the inner capsid protein VP6, RVs are classified into twelve distinct species (RVA–RVL) [7]. Previous studies detected RVA–RVC, RVE, and RVH in pigs; among them, RoVA is considered the most common pathogenic RV species associated with infectious diarrhea [8,9,10].

RVs are non-enveloped, double-stranded RNA viruses with an 11-segmented genome [6]. The two outer capsid proteins (VP7 and VP4) stimulate production of neutralizing antibodies and form the basis for the G (Glycoprotein) and P (Protease-sensitive) dual (G/P) genotyping system for RVA strains [11,12,13]. This dual genotyping system is often used for RVA classification, and is updated continuously by the Rotavirus Classification Working Group. According to this classification system, 42 G- and 58 P genotypes have been described globally [6,14] (https://rega.kuleuven.be/cev/viralmetagenomics/virus-classification/rcwg, accessed on 6 October 2022). The diversity of G/P combinations is an important determinant of immune protection, and is highly relevant to vaccine development [14,15]. For this reason, continuous surveillance of circulating RVA strains is important for improving regional epidemiological information and for updating novel vaccine strains. All porcine RV vaccines in current use are based on porcine group A rotavirus (PoRVA) strains. Despite the use of commercial PoRVA vaccines, RVA infections and associated diarrhea are still relatively common [6]. Until recently, 12 G- (G1–G6, G8–G12, and G26) and 13 P- (P[1], P[5]–P[8], P[11], P[13], P[19], P[23], P[26], P[27], P[32], and P[34]) genotypes of PoRVA have been reported on pig farms worldwide [16]. Furthermore, the prevalent G and P genotype combinations vary and change depending on the geographical place and the time [6,15,17,18,19,20].

Thus, molecular epidemiological data on Korean PoRVA strains circulating on pig farms are needed to determine the prevalence of the G and P genotypes, and to update the vaccine strains. Here, we investigated the G and P genotypes of the PoRVA strains prevalent in pig farms in South Korea. Two epidemic viruses were isolated for use as potential new vaccine strains.

## 2. Materials and Methods

### 2.1. Sample Collection and Preparation

A total of 377 porcine diarrheic fecal samples were collected from piglets (aged < 30 days) on farms across eight provinces (Gyeonggi, Gangwon, Chungbuk, Chungnam, Gyeongbuk, Gyeongnam, Jeonbuk, and Jeonnam) in Republic of Korea between 2014 and 2018. The number of samples and year of collection were as follows: 2014 (*n* = 31), 2015 (*n* = 7), 2016 (*n* = 24), 2017 (*n* = 290), and 2018 (*n* = 25). These samples were diluted 1:3 with phosphate-buffered saline (PBS) and centrifuged at 3000× *g* for 20 min. The supernatants were filtered through a 0.22 μm-pore-syringe filter (Minisart^®^ syringe filter, Sartorius, Germany) and stored at −80 °C until further analysis.

### 2.2. PCR and Sequencing

Total RNA was extracted from each supernatant sample using the RNeasy mini kit (Qiagen Inc., Germantown, MD, USA), and eluted with 50 μL of elution buffer. Total RNA (200 ng) was reverse-transcribed using the HelixCript^TM^ easy cDNA synthesis kit (Nonohelix Co., Ltd., Daejeon, Republic of Korea). All synthesized cDNA samples were used immediately for PCR amplification or stored at −20 °C until use. To detect the rotaviral RNA in fecal samples, PCR assays were performed using primer pairs specific for partial regions of the PoRVA outer capsid genes VP7 and VP4 [17]. PCR assays were carried out using the AccuPower^®^ ProFi Taq PCR PreMix kit (Bioneer Inc., Daejeon, Republic of Korea) under the following conditions: initial denaturation at 95 °C for 3 min; followed by 40 cycles of denaturation at 95 °C for 30 s, annealing at 52 °C (VP7) or 55 °C (VP4) for 30 s, extension at 68 °C for 1 min, and a final elongation step at 68 °C for 5 min. Subsequently, all amplified products were analyzed by electrophoresis on 1.5% agarose gels containing ethidium bromide (EtBr), and visualized under an ultraviolet image analyzer. All purified PCR products were sequenced using the sequencing service of a professional molecular analysis facility (CosmoGENTECH Inc., Seoul, Republic of Korea). After sequencing, the nucleotide sequences were compared with reference data available in the GenBank database using the Basic Local Alignment Search Tool (BLAST, http://blast.ncbi.nlm.nih.gov/Blast.cgi, accessed on 25 March 2022) program.

### 2.3. Genotyping and Phylogenetic Analysis

The genotypes for the two major genome segments (VP7 and VP4) were determined using nucleotide BLAST and a web-based RotaC genotyping tool; http://viprbrc.org, accessed on 14 April 2022, [21]. Multiple sequence alignment (including reference sequences) was performed using CLUSTAL X (v2.1) (http://www.clustal.org/clustal2). Phylogenetic trees based on partial nucleotide sequences of VP7 and VP4 were constructed using the maximum-likelihood (ML) method in Molecular Evolutionary Genetics Analysis X (MEGA X) software (v10.2.6) [22], with nucleotide distance (*p*-distance) and 1000 replications used for bootstrap analysis. Before each phylogenetic analysis, the model of nucleotide substitution that best fitted the data was identified using the “find best model” function in MEGA X. The PoRVA reference strains were isolated in Asia, America, and Europe, including the 2006–2018 strains detected in Republic of Korea. Complete VP7 and VP4 gene sequences of the PoRVA reference strains were obtained from the National Center for Biotechnology Information (NCBI) GenBank database. 

### 2.4. Virus Isolation and Passage

PoRVA was isolated in TF-104 cells (a cloned derivative of African green monkey kidney epithelial MA-104 cells) as described previously, with some modifications [12,17]. Briefly, TF-104 cells were grown in 6-well plates in α-Minimum Essential Medium (α-MEM, Gibco^TM^, Waltham, MA, USA) supplemented with 10% fetal bovine serum (GenDEPOT, Katy, TX, USA), 7.5% sodium bicarbonate (Sigma-Aldrich, New York, NY, USA), 0.5% tryptose phosphate broth (Sigma-Aldrich, New York, NY, USA), and antibiotic–antimycotic solutions (100×, Gibco^TM^, Waltham, MA, USA). Confluent TF-104 cell monolayers were washed with PBS and used for virus inoculation. Before inoculation, filter-sterilized PoRVA-positive samples were treated with trypsin solution (2.5%; Gibco^TM^, Waltham, MA, USA; final concentration, 10 μg/mL) at 37 °C for 60 min. Next, a PoRVA-positive inoculum was added to the TF-104 cell monolayer. After incubating at 37 °C for 1 to 2 h, the inoculum was removed and 2 mL of virus growth medium [α-MEM supplemented with 7.5% sodium bicarbonate, 0.5% tryptose phosphate broth, antibiotic–antimycotic solution, and 5 μg/mL trypsin] was added. The inoculated cell plates were incubated at 37 °C under 5% CO_2_ and examined daily for significant cytopathic effects (CPE). When CPE appeared in more than 80% of cells (approximately 3 days after inoculation), the plates were frozen at −70 °C and thawed twice. The culture supernatants were used as viral seed stocks for the next blind passage. The isolated PoRVA strains were confirmed by PCR and in a direct immunofluorescence (IF) assay (VDPro^®^ Rota FA reagent, Median Diagnostics, Chuncheon, Republic of Korea). If the CPE and PCR results were negative after five blind passages, virus isolation was considered negative. The culture scale was increased gradually until PoRVA strains could be cultured in T-75 flasks.

## 3. Results

### 3.1. Detection of Porcine Rotavirus Antigens

Between 2014 and 2018, 377 diarrheic fecal samples collected from pig-producing regions across South Korea were tested for the PoRVA antigen; of these, 53 (14.1%) were positive (Table 1). According to geographical region, the PoRVA-positive rates over the 5 years were as follows: 15.3% (8/52) in Gyeonggi (GG); 23.6% (13/55) in Gangwon (GW); 11.7% (6/51) in Chungbuk (CB); 24.5% (13/53) in Chungnam (CN); 5.7% (3/52) in Gyeongbuk (GB); 7.5% (4/54) in Gyeongnam (GN); 11.3% (6/54) in Jeonbuk (JB); and 0% (0/6) in Jeonnam (JN) (Figure 1). The yearly PoRVA-positive rates were 29.0% (9/31) in 2014, 42.8% (3/7) in 2015, 54.2% (13/24) in 2016, 9.0% (26/290) in 2017, and 8.0% (2/25) in 2018 (Table 2).

### 3.2. Korean PoRVA Genotypes

To determine RV G and P genotypes, 53 PoRVA-positive fecal samples from pigs with suspected rotaviral enteritis were tested by partial nucleotide sequencing and phylogenetic analysis of the VP7 and VP4 genes (Table 1).

As a result, we identified 15 different G and P genotype combinations. A previous study of the distribution of G and P combinations of PoRVA strains during the year 2006–2007 [17] showed that G5P[7] was the single most prevalent combination (64.3%) (Figure 2A). However, of the 53 Korean PoRVA strains identified in this study, the G3, G4, G5, G9, G11, and G26 genotypes were prevalent. Among them, G4 and G9 were the predominant VP7 genotypes, each accounting for 39.6% (21/53) of strains (Figure 2B). The prevalent VP4 genotypes were P[7] (33.9%, 18/53), P[6] (30.2%, 16/53), P[23] (17.0%, 9/53), P[13] (15.1%, 8/53), and P[26] (3.8%, 2/53) (Figure 2B). Overall, the prevalent G and P genotype combinations were G4P[6] (18.9%, 10/53), G9P[7] (17.0%, 9/53), G9P[23] (11.3%, 6/53), G3P[7] (9.4%, 5/53), and G9P[13] (7.5%, 4/53). The remaining ten genotype combinations represented < 5.7% (Figure 2B).

### 3.3. Prevalence of the Korean PoRVA G and P Genotypes over 5 Years

The G4 (2014–2018) and G9 (2014–2017) genotypes were detected continuously across the country (the exceptions were G4 and G9 in CB province) every year (Table 2). In the case of PoRVA-positive G3, G5, G11, and G26 genotypes, a small number were identified only in some provinces between 2016 and 2017. This result shows that the epidemic PoRVA G and P genotypes, which are currently a problem for domestic pig farms in South Korea, are G4 and G9. Meanwhile, the P[6] and P[7] genotypes were identified every year between 2014 and 2017 in most regions (Table 2). These results suggest that the most prevalent P genotypes of PoRVA on pig farms in South Korea are P[6] and P[7].

### 3.4. Sequence and Phylogenetic Analysis of the Partial VP7 Gene

A comparison of the nucleotide sequences of a partial VP7 gene sequence (a 935 fragment) from the 53 Korean PoRVA strains examined in this study with representative PoRVA strains (including commercial vaccine strains: Gottfried (G4), OSU (G5), and A2 (G9)) was performed (data not shown). Twenty-one of the 53 Korean PoRVA strains showed 88.9–95.7% identity with the G9 genotype PoRVA vaccine strain A2 from the USA (2004, AB180971). In addition, 21 Korean PoRVA G4 genotype strains showed low identity (85.0–87.1%) at the nucleotide level with the G4 genotype PoRVA vaccine strain Gottfried from the USA (1975, MT025916).

A comparison of the full-length nucleotide and deduced amino acid sequences of the VP7 gene of the RVA/pig/wt/KOR/GN-108/2017/G26 strain (accession no. OP3211073) detected in this study with reference strains from GenBank revealed that the GN-108 strain shares the closest similarity to the RVA/pig-wt/JPN/TJ4-1 strain (AB605258) from Japan both at the nucleotide (96.1%) and amino acid (97.7%) level. In addition, the GN-108 strain was homologous (86.5–93.8% at the nucleotide level and 90.1–97.4% at the amino acid level) with six strains (KT277523–KT277528) of the G26 genotype isolated from the Tripura and Assam regions in India.

Phylogenetic analysis of the partial nucleotide sequences of the VP7 gene from the 53 Korean PoRVA strains, and the selected reference strains from GenBank, was performed using the maximum-likelihood method in MEGA X (Figure 3). The phylogenetic tree for VP7 genes revealed that 53 Korean PoRVAs belonged to the G3, G4, G5, G9, G11, and G23 types. In particular, the 21 Korean PoRVAs belonging to the G9 type were closely related to the Japan strain JP29-6 and the Thailand strain CMP003, and were also included in the same type as vaccine strains Porcilis, 10, A1, and A2. Twenty-one Korean PoRVAs belonged to the G4 type, which first occurred in South Korea and several other countries (USA, China, Thailand, and Japan), along with the vaccine strain Gottfried. In addition, six Korean PoRVA strains belonged to the G3 type, which is the first occurrence in South Korea; this type also included the China strains TA-4-2 and HLJheb-1, and the Taiwan strain CMP213. The accession numbers of the reference sequences reported worldwide, including the Korean strains, are shown in Figure 3 and Figure 4.

### 3.5. Sequence and Phylogenetic Analyses of a Partial VP4 Gene Sequence

A comparison of the partial nucleotide sequence of the VP4 gene (an 816 fragment) from the 53 Korean PoRVA strains identified in this study and representative PoRVA strains (including commercial vaccine strains Gottfried (P[6]), and A2 (P[7])) was performed. Eighteen of the 53 Korean PoRVA strains were 90.4–96.9% identical to the P[7] genotype PoRVA vaccine strain A2 from the USA (2004, AB180977). Sixteen Korean PoRVA P[6] genotype strains showed low nucleotide identity (79.1–81.9%) with the P[6] genotype PoRVA vaccine strain Gottfried from the USA (1990, M33516). Phylogenetic analysis for the partial nucleotide sequences of the VP4 gene from the 53 Korean PoRVA strains and selected reference strains from GenBank was also performed using the maximum-likelihood method in MEGA X (Figure 4).

The phylogenetic tree for VP4 revealed that 53 Korean PoRVAs belonged to five types (P[6], P[7], P[13], P[23], and P[26]), and that the P[7] type, which contains the 18 Korean PoRVAs, was closely related to the vaccine strains Porcilis, 10, A1, and A2 strains. Eight Korean PoRVAs, the China strain TM-a, and the USA strain A34 belonged to the P[23] type, whereas the P[13] type contained Korean PoRVAs that were first detected in South Korea and belonged to the same type as strains from various countries (Canada, USA, Japan, and Thailand).

### 3.6. Isolation of the Novel Korean PoRVA Strains from TF-104 Cells

Of the 53 PoRVA-positive fecal samples, seven were selected for viral isolation in the TF-104 cell line using the blind passage strategy. The seven samples (G9P[7]:JB-57 (2017), CN-67 (2017), G4P[6]:GB-68 (2017), CN-119 (2017), CN-145 (2017), G4P[13]:CN-202 (2017), and GN-368 (2018)) were selected based on the predominant G and P genotypes among relatively recent samples (2017 to 2018). The selected PoRVA-positive samples were diluted 10-fold and inoculated into TF-104 cells. The TF-104 cells were monitored for CPE after five continuous blind passages. Specific CPEs, characterized by the rounding and detachment of cell clusters, were observed in 10-passage cell cultures after 24–48 h. Finally, two Korean novel PoRVAs (CN-67:G9P[7] and CN-202:G4P[13]) adapted successfully to the TF-104 cells and propagated rapidly. These two novel viruses were confirmed by detecting PoRVA antigens by IFA using RVA VP6-specific MAbs (Figure 5).

## 4. Discussion

Previously, it has been reported that the prevalence of PoRVA in Korean pigs identified on 143 farms across six provinces between 2006 and 2007 was 20.6% (98/475) [17]. The G5 type accounted for 67.3%, followed by G8 (7.4%), G9 (9.2%), and unknown (6.1%) [17]. By contrast, the P7 type was the most prevalent at 92.9%; the rest had a very low prevalence (2% for P[1] and P[23], and 3.1% for unknown types) [17]. In addition, sequence and phylogenetic analyses of PoRVA strains identified between 2006 and 2007 revealed that the predominant combinations were G5P[7] (64.3%), G8P[7] (16.3%), G9P[7] (7.1%), G9P[23] (2%), and G8P[1] (1%) [17].

However, 10 years later, the present study shows that various G types (G3, G4, G5, G9, G11, and G26) and P types (P[6], P[7], P[13], P[23], and P[26]) remain prevalent in South Korea. We found that the most common combination of G and P genotypes had changed: G5P[7] had very low detection rates (1.9%), and G8P[7] and G8P[1] were not detected at all, whereas the detection rates of G9P[7] and G9P[23] were 17.0% and 11.3%, respectively. Since the newly emerged G4 genotype is the most prevalent, the detection rate of complexes containing P types [6], [7], [13], [23], and [26] is close to 40%, making them the most likely cause of diarrheal disease on Korean pig farms. This situation may make it more difficult to eradicate rotaviruses from pig farms, and increases the risk of severe and acute dehydrating diarrhea in piglets. Considering that the genetic diversity of PoRVA in South Korea has gradually increased, economic losses caused by the viruses in the Korean swine industry should not be underestimated.

A novel PoRVA G26 genotype caused four different diarrheal outbreaks in Japan in 2009–2010, which affected almost all suckling pigs born to 20–30% of lactating sows [23]. Recently, a novel G26P[13] genotype was identified in a PoRVA-infected pig population in the Tripura and Assam regions of India [24]. A comparison of the nucleotide and amino acid sequences of the VP7 gene of the GN-108 (G26 genotype) strain that was detected in the present study showed great similarity with the TJ4-1 strain detected in Japan and six strains (TR/RV/SW21, TR/RV/SW25, TR/RV/SW49, TR/RV/SW51, TR/RV/SW76, and AS/RV/SW90) detected in India. Although the mortality and morbidity caused by the G26 genotype found in pigs from GN province in South Korea in 2017 were not investigated in this study, this new type is expected to cause piglet diarrheal disease similar to that observed in Japan and India. Therefore, further characterization is now needed to determine the pathogenic potential of the G26 genotype.

In particular, the novel G4 genotype occurred in six regions (the exceptions were CB and JN). The G4 genotype first occurred in GW in 2014–2015, in GG in 2016, in JB, GB, and CN in 2017, and in GN in 2018. A new G4 genotype was generated through a combination of P[6] and P[23] on pig farms in 2014; it then recombined with P[7] and P[13] in 2017. As such, the G4 genotype combined with various P genotypes from 2014 to 2018 and comprises a great proportion (23.1–100% per year) of PoRVA-positive samples. Thus, the importance of the G4 genotype is based on the prediction that it will combine with new P genotypes in the future and cause economic damage to pig farms; therefore, continuous monitoring is required. 

Currently, the domestic bivalent vaccine (containing the G5P[7]-A1 and G9P[7]-10 strains), along with imported vaccines such as the ProSystem RCE vaccine, is used to protect against PoRVA in South Korea [14]. Although the imported vaccine (ProSystem RCE vaccine) contains the G4 strain, the Korea Animal Medicine Association in 2021 reported that approximately 83% of the total PoRVA vaccine sales were made by Korean vaccine companies. Therefore, to prevent a new epidemic of the G4 genotype, imported vaccines containing the G4 genotype should be used as much as possible. In addition, there is a need for Korean vaccine companies to develop a PoRVA vaccine type containing the G4 strain.

To isolate PoRVA from fecal samples of diarrheic piglets, previous studies attempted inoculation into MA-104 [12,14,25] and TF-104 [17] cells. Another study reported that the detection rate of PoRVA VP7 sequences was the highest in 30-day-old pigs (67%) [26]. In general, the detection rate of the PoRVA gene in pig diarrheic feces is high, but the virus isolation rate is very low. To increase the rate of rotavirus isolation, we used diarrheal samples from piglets under 30 days of age [26], as well as suitable cells (TF-104) [17] and a trypsin concentration of 10 μg/mL [12,17]. It is predicted that the improved isolation method for PoRVA will aid in increasing the virus isolation rate in future studies.

In this study, we tried to isolate G4P[6] and G9P[7], which were the most prevalent among the 53 PoRVAs. CN-67 (G9P[7]), which was isolated from diarrheic fecal samples from 30-day-old pigs, successfully adapted to TF-104 cells, but G4P[6] failed to adapt and could not be isolated. Instead, the CN-202 (G4P[13]) strain successfully adapted to TF-104 cells. In future studies, two novel Korean PoRVA (CN-67(G9P[7]) and CN-202 (G4P[13]) strains isolated from diarrheal samples will be subjected to virulence testing in piglets and to continuous passage in TF-104 cells in order to confirm their potential as attenuated vaccine candidates.

## 5. Conclusions

The data presented herein provide valuable epidemiological information and genetic characteristics of the various field-type strains of PoRVA circulating on pig farms in South Korea. A novel G4 genotype vaccine targeting the current Korean circulating strains may be required for more effective control of PoRVA.

## Figures and Tables

**Figure 1 viruses-14-02522-f001:**
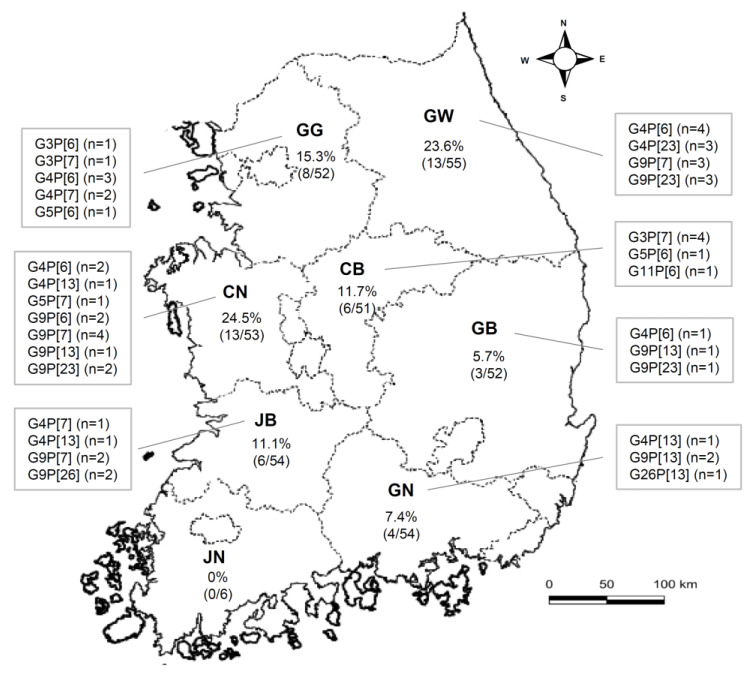
Regional prevalence of 53 PoRVA strains detected in Korean pigs between 2014 and 2018.

**Figure 2 viruses-14-02522-f002:**
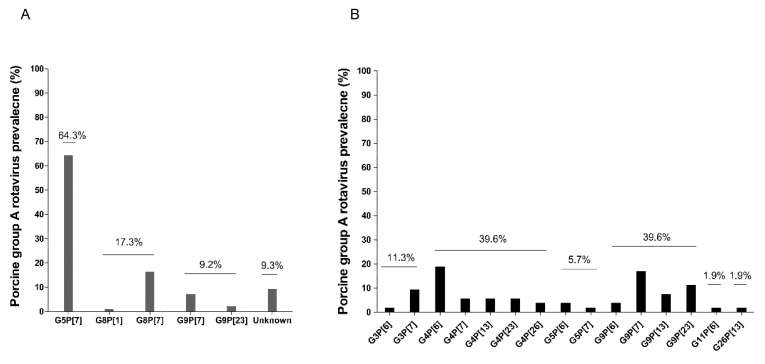
Comparison of the distribution of the G and P combinations of PoRVA strains that have circulated in pigs in South Korea. Distribution of the G/P combinations of PoRVA strains during 2006–2007 (*n* = 98) [17] (**A**) and 2014–2018 (*n* = 53) (**B**).

**Figure 3 viruses-14-02522-f003:**
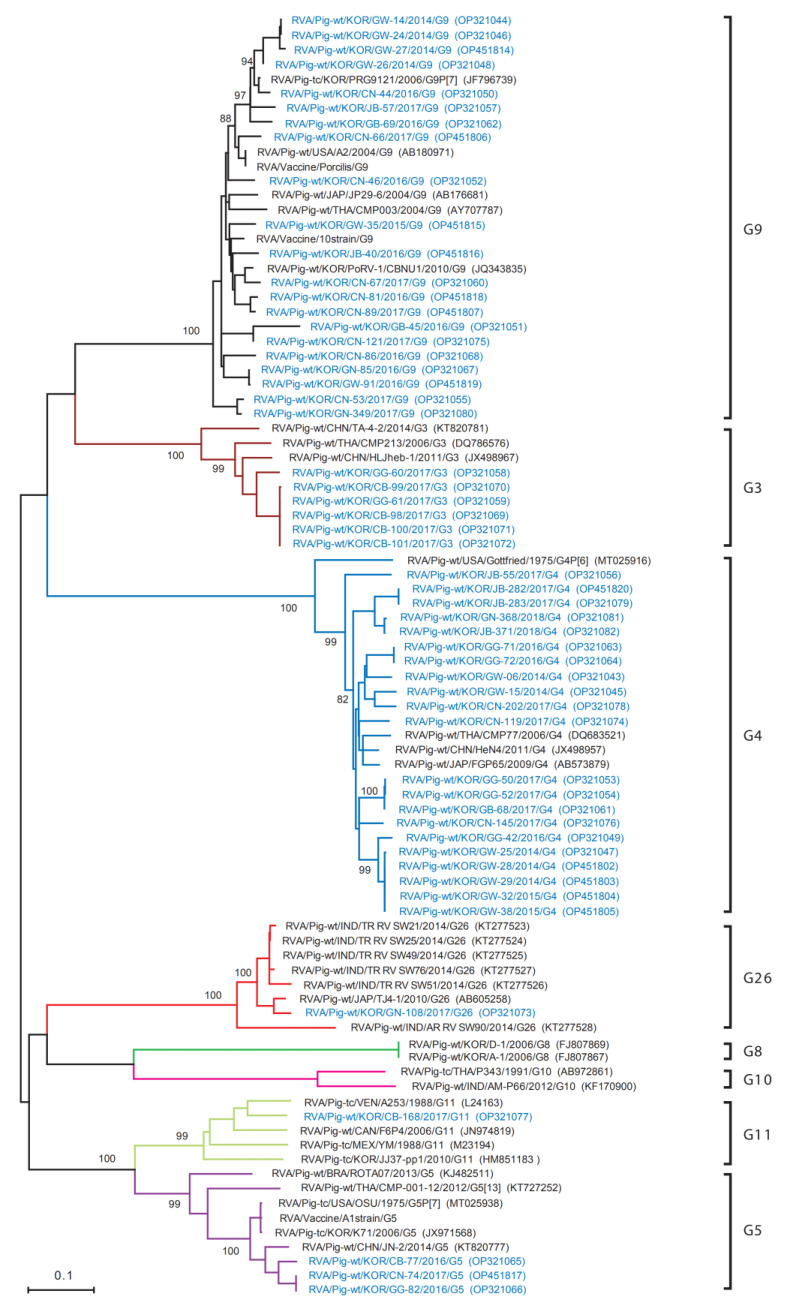
Phylogenetic analysis based on a partial nucleotide sequence of the VP7 gene from 53 PoRVA antigen-positive samples and selected reference strains. The phylogenetic tree was reconstructed using the maximum-likelihood (ML) method (based on the Tamura 3-parameter model) in the MEGA X program, and tested using 1000 bootstrap values. The 53 Korean PoRVA isolates identified in this study are denoted by blue letters.

**Figure 4 viruses-14-02522-f004:**
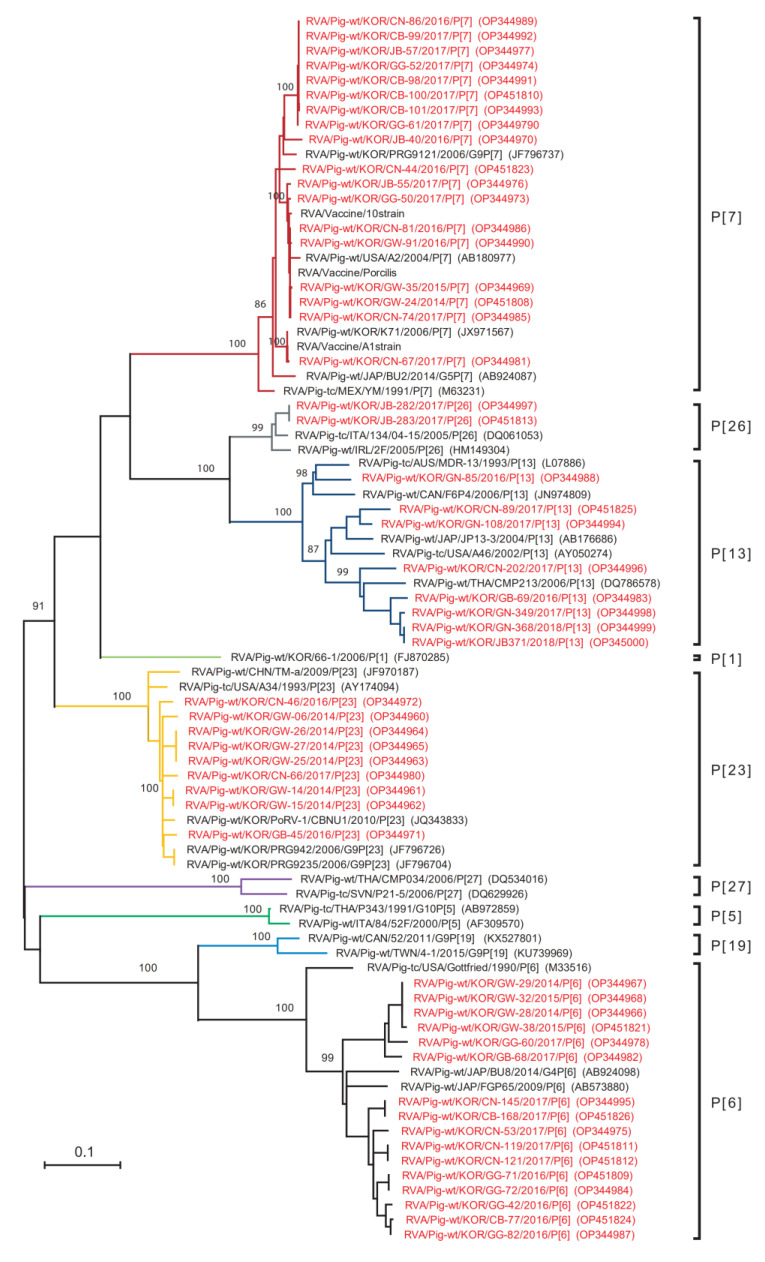
Phylogenetic analysis based on a partial nucleotide sequence of the VP4 gene from 53 PoRVA antigen-positive samples and selected reference strains. The phylogenetic tree was reconstructed based on the HKY (Hasegawa-Kishino-Yano) model in the MEGA X program, with 1000 bootstrap values. The 53 Korean PoRVA isolates identified in this study are denoted by red letters.

**Figure 5 viruses-14-02522-f005:**
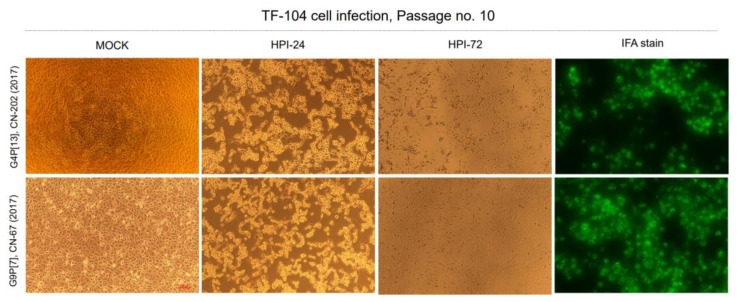
Isolation and detection of the PoRVA CN-202 (G4P[13]) and CN-67 (G9P[7]) strains in TF-104 cells. TF-104 cells were seeded in T-75 flasks and infected with 10th passage virus at a multiplicity of infection (MOI) of 0.1. Specific cytopathic effects were observed under a microscope at 24–72 h postinfection (hpi) (Mag × 100). The immunofluorescence assay results for the PoRVA CN-202 (G4P[13]) and CN-67 (G9P[7]) strains (passage 10) in infected TF-104 cells at 20 hpi are shown (Mag × 100).

**Table 1 viruses-14-02522-t001:** G and P genotyping data for 53 Korean PoRVA-positive samples obtained between 2014 and 2018.

Genotype	G3	G4	G5	G9	G11	G26	Total
P[6]	1	10	2	2	1		16 (30.2%)
P[7]	5	3	1	9			18 (33.9%)
P[13]		3		4		1	8 (15.1%)
P[23]		3		6			9 (17.0%)
P[26]		2					2 (3.8%)
Total	6 (11.3%)	21 (39.6%)	3 (5.7%)	21 (39.4%)	1 (1.9%)	1 (1.9%)	

**Table 2 viruses-14-02522-t002:** G and P genotyping data for 53 Korean PoRVA-positive samples according to geographical region (2014–2018).

Years	G and P Genotyping by Regions							
	GG * (8/52) **	GW (13/55)	CB (6/51)	CN (13/53)	GB (3/52)	GN (4/54)	JB (6/54)	JN (0/6)
2014 (9/31) **		G4P[6] (*n* = 2) G4P[23] (*n* = 3) G9P[7] (*n* = 1) G9P[23] (*n* = 3)						
2015 (3/7)		G4P[6] (*n* = 2) G9P[7] (*n* = 1)						
2016 (13/24)	G4P[6] (*n* = 3) G5P[6] (*n* = 1)	G9P[7] (*n* = 1)	G5P[6] (*n* = 1)	G9P[7] (*n* = 3) G9P[23] (*n* = 1)	G9P[13] (*n* = 1) G9P[23] (*n* = 1)	G9P[13] (*n* = 1)	G9P[7] (*n* = 1)	
2017 (26/290)	G3P[6] (*n* = 1) G3P[7] (*n* = 1) G4P[7] (*n* = 2)		G3P[7] (*n* = 4) G11P[6] (*n* = 1)	G4P[6] (*n* = 2) G4P[13] (*n* = 1) G5P[7] (*n* = 1) G9P[6] (*n* = 2) G9P[7] (*n* = 1) G9P[13] (*n* = 1) G9P[23] (*n* = 1)	G4P[6] (*n* = 1)	G9P[13] (*n* = 1) G26P[13] (*n* = 1)	G4P[7] (*n* = 1) G9P[26] (*n* = 2) G9P[7] (*n* = 1)	
2018 (2/25)						G4P[13] (*n* = 1)	G4P[13] (*n* = 1)	

* GG: Gyeonggi; GW: Gangwon; CB: Chungbuk; CN: Chungnam; GB: Gyeongbuk; GN: Gyeongnam; JB: Jeonbuk; JN: Jeonnam. ** Number of PoRVA-positive samples/number of tested samples.

## Data Availability

The nucleotide sequences of the 2014–2018 viruses obtained in this study were submitted to the GenBank database under accession numbers OP321043-OP321082, OP451802-OP451807, OP451814-OP451820 for the VP7 gene, and OP344960-OP345000, OP451808-OP451813, OP451821-OP451826 for the VP4 gene.

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
