# Peer review of "Genetic Diversity of Porcine Group A Rotavirus Strains from Pigs in South Korea"

_viruses, 2022, doi:10.3390/v14112522_

Round 1

Reviewer 1 Report

Summary:

Group A porcine rotaviruses (PRoVAs) cause acute gastroenteritis and dehydrating diarrhea in piglets, making it a significant economic concern. In this study, Park et al. analyzed the VP7 and VP4 genes from PRoVAs isolated from 53 domestic pig diarrheic fecal samples in South Korean farms between the years 2014 and 2018. The results of this study identified six G-genotypes and five P-genotypes circulating in South Korea, and a marked shift in predominating genotypes when compared to studies from 2006-2007. Notably, G4 and G9 genotypes as well as P[7] and P[6] genotypes were most commonly identified during the time frame of this study, which may not be fully covered by the genotype vaccines currently used in South Korea. These results raise the possibility that novel PRoVA vaccine strains need to be developed to address the changing landscape of rotavirus diversity on South Korean pig farms. Overall, the results of this study are clear and easy to read.

Major Comments:

11)      While I think the information in table 2 is informative as laid out, it is visually confusing to sort out genotype circulation trends as described in the text. As either an additional figure or supplemental figure, I would recommend arranging these data to show G and P genotype distribution by year.

Minor Comments:

22)      As an addendum to comment 1, there is text in lines 180-181 that details the decrease in the number of P[23] genotyped samples from 2014 onwards. While I believe the data in table 2 can be used to highlight common persisting genotypes isolated across different regions, I think the overall dataset is ultimately too small to write definitively about rising and falling trends and would caution against doing so.

33)      Regarding Figure 1, please include numbers that show the total number of respective samples analyzed in Figure 1A and Figure 1B.

44)      The abstract of this work highlights that G5 and G9 genotype vaccines are currently used in South Korea against PRoVA infection, which may not fully protect against the novel G9P[7] and G4P[13] genotypes identified in this study, among others. This strikes me as a very important observation, and I was surprised to see this was not touched on in detail in the discussion. Please amend this section to include these considerations.      

Author Response

Review 1

Group A porcine rotaviruses (PRoVAs) cause acute gastroenteritis and dehydrating diarrhea in piglets, making it a significant economic concern. In this study, Park et al. analyzed the VP7 and VP4 genes from PRoVAs isolated from 53 domestic pig diarrheic fecal samples in South Korean farms between the years 2014 and 2018. The results of this study identified six G-genotypes and five P-genotypes circulating in South Korea, and a marked shift in predominating genotypes when compared to studies from 2006-2007. Notably, G4 and G9 genotypes as well as P[7] and P[6] genotypes were most commonly identified during the time frame of this study, which may not be fully covered by the genotype vaccines currently used in South Korea. These results raise the possibility that novel PRoVA vaccine strains need to be developed to address the changing landscape of rotavirus diversity on South Korean pig farms. Overall, the results of this study are clear and easy to read.

Major Comments:

Comment 1: While I think the information in table 2 is informative as laid out, it is visually confusing to sort out genotype circulation trends as described in the text. As either an additional figure or supplemental figure, I would recommend arranging these data to show G and P genotype distribution by year.

Answer: In accordance with the reviewer’s comment, we have revised Table 2 and added a new Figure 1.

Minor Comments:

Comment 2:  As an addendum to comment 1, there is text in lines 180-181 that details the decrease in the number of P[23] genotyped samples from 2014 onwards. While I believe the data in table 2 can be used to highlight common persisting genotypes isolated across different regions, I think the overall dataset is ultimately too small to write definitively about rising and falling trends and would caution against doing so.

Answer: We have removed the sentence from the revised manuscript.

Comment 3:  Regarding Figure 1, please include numbers that show the total number of respective samples analyzed in Figure 1A and Figure 1B.

Answer: We have changed the original Figure 1 to Figure 2, and revised Figure 1A and 1B.

“Figure 2. Comparison of the distribution of the G and P combinations of PoRVA strains circulated in pigs in South Korea. Distribution of the G/P combinations of PoRVA strains during 2006–2007 (n=98) [17] (A) and 2014–2018 (n=53) (B).”

Comment 4: The abstract of this work highlights that G5 and G9 genotype vaccines are currently used in South Korea against PRoVA infection, which may not fully protect against the novel G9P[7] and G4P[13] genotypes identified in this study, among others. This strikes me as a very important observation, and I was surprised to see this was not touched on in detail in the discussion. Please amend this section to include these considerations.      

Answer: In accordance with the reviewer’s comment, we have now mentioned this in the Discussion (lines 244—250, 260–267, 279–288, 295–307).

Reviewer 2 Report

The manuscript investigated the prevalence of predominant circulating genotypes by analyzing the VP7 and VP4 genes of PRoVAs isolated between 2014 and 2018 from domestic pigs in South Korea. The resulting data provide valuable epidemiological information and genetic characteristics of various field-type strains of PRoVA circulating on pig farms in South Korea, which are helpful for novel vaccine strain update.

Major concerns:

Line 164-169, “most prevalent” should be correctly used in these comparisons. Similarly, “ the most predominant combinations ...” in Line 274.

Line 195-196 and 206-207, the results in Figure 2 and 3 should be described in detail, not merely a description that the results are shown in Figures.

Line 288-293, the reviewer don’t quite understand why these two references are described in parallel, especially, “whereas another reported that the detection rate of PRoVA VP7 sequences was highest in 30-day-old pigs (67%), followed by suckling pigs (43%), lactating sows (17%), and 120-day-old pigs (7%)”. Does the authors mean successful isolation of two novel Korean strains and adaptation to TF-104 cells is attributed to the selection of 30-day-old pigs in which the highest PRoVA prevalence was detected? If so, related evidences and references should be provided.

Author Response

Review 2

The manuscript investigated the prevalence of predominant circulating genotypes by analyzing the VP7 and VP4 genes of PRoVAs isolated between 2014 and 2018 from domestic pigs in South Korea. The resulting data provide valuable epidemiological information and genetic characteristics of various field-type strains of PRoVA circulating on pig farms in South Korea, which are helpful for novel vaccine strain update.

Major concerns:

Comment 1: Line 164-169, “most prevalent” should be correctly used in these comparisons. Similarly, “ the most predominant combinations ...” in Line 274.

Answer: We have revised the text accordingly (lines 158–164, 271–272).

Comment 2: Line 195-196 and 206-207, the results in Figure 2 and 3 should be described in detail, not merely a description that the results are shown in Figures.

Answer: Thank you. We have revised the sentence accordingly (lines 195–206, 215–223).

Comment 3: Line 288-293, the reviewer don’t quite understand why these two references are described in parallel, especially, “whereas another reported that the detection rate of PRoVA VP7 sequences was highest in 30-day-old pigs (67%), followed by suckling pigs (43%), lactating sows (17%), and 120-day-old pigs (7%)”. Does the authors mean successful isolation of two novel Korean strains and adaptation to TF-104 cells is attributed to the selection of 30-day-old pigs in which the highest PRoVA prevalence was detected? If so, related evidences and references should be provided.

Answer: Thank you. We have revised the sentences accordingly (lines 295–307).

Reviewer 3 Report

General comments:

The manuscript describes the epidemiology and molecular characterization of RVA in pigs in South Korea. The authors demonstrated the evolution of this virus in the country in relation to what was observed 10 years earlier. These data reinforce the need for continuous monitoring of these infections. In this way, I believe that the work deserves publication. However, some aspects of the manuscript need to be improved.

 Specific comments:

PRoVAs: This is not common terminology for rotavirus. Porcine rotavirus generally does not have an acronym; however, if you want to use it, I recommend switching to PoRVA throughout the manuscript.

The correct acronym for rotaviruses is RVs and not RoVs. Please correct the entire manuscript.

 Introduction:

Page 1

Lines 35-36: RoVs are classified into ten distinct serogroups (RoVA–RoVJ).

-The correct terminology is no longer serogroup but species. Please correct.

-Recently 2 new species were described. See Johne, R.; Tausch, S.H.; Grützke, J.; Falkenhagen, A.; Patzina-Mehling, C.; Beer, M.; Hoper, D.; Ulrich, R.G. Distantly Related Rotaviruses in Common Shrews, Germany, 2004–2014. Emerge info Dis. 2019, 25, 2310–2314.

Line 37-38: RoVA–RoVC, RoVE, and RoVH - Correct acronym for RVA, RVC, RVE, and RVH.

Lines 45-46: According to this classification system, 35 G- and 50 P- genotypes have been described globally - Currently, 42 G- and 58 P-genotypes are described. https://rega.kuleuven.be/cev/viralmetagenomics/virus-classification/rcwg

 Results:

Page 3, lines 130-137: The authors comment on positivity by geographic region over 5 years and annual positivity rates. However, these data are not clearly shown in the tables. The tables are confusing. It is unclear how many samples were obtained by region/year.

Perhaps it would be better if Table 1 presented the overall results showing the number of tested/positive samples by year and region.

In Table 2, the genotyping result could be introduced. And maybe a graph showing the regions and genotypes identified by region/year.

Page 6, lines 163-169:

Please rephrase. Based on the data presented, although genotypes G3, G5, G11 and G26 were detected, they were not "most prevalent". The G4 and G9 genotypes were the most prevalent. The same occurred with respect to genotypes P[13], P[23] and P[26].

 Discussion:

The discussion needs to be greatly improved. The text is repetitive. It is not necessary to present the detection percentages for each genotype in different countries. The authors have already presented, in the introduction, which RVA genotypes are most prevalent in the world.

It would be more fruitful to focus the discussion on the evolution of epidemiology in South Korea and the implications of this in relation to the development of measures to prevent and control these infections.

Author Response

Review 3

The manuscript describes the epidemiology and molecular characterization of RVA in pigs in South Korea. The authors demonstrated the evolution of this virus in the country in relation to what was observed 10 years earlier. These data reinforce the need for continuous monitoring of these infections. In this way, I believe that the work deserves publication. However, some aspects of the manuscript need to be improved.

Specific Comments:

Comment 1: PRoVAs: This is not common terminology for rotavirus. Porcine rotavirus generally does not have an acronym; however, if you want to use it, I recommend switching to PoRVA throughout the manuscript. The correct acronym for rotaviruses is RVs and not RoVs. Please correct the entire manuscript.

Answer: Thank you. We have made the change as suggested.

Introduction:

Comment 1: Page 1 Lines 35-36: RoVs are classified into ten distinct serogroups (RoVA–RoVJ). The correct terminology is no longer serogroup but species. Please correct. Recently 2 new species were described. See Johne, R.; Tausch, S.H.; Grützke, J.; Falkenhagen, A.; Patzina-Mehling, C.; Beer, M.; Hoper, D.; Ulrich, R.G. Distantly Related Rotaviruses in Common Shrews, Germany, 2004–2014. Emerge info Dis. 2019, 25, 2310–2314.

Answer: Thank you. We have added the two new serogroups (RVK and RVL) (line 37). We have also added the reference (Johne et al., 2019).

Comment 2: Line 37-38: RoVA–RoVC, RoVE, and RoVH - Correct acronym for RVA, RVC, RVE, and RVH.

Answer: We have made the suggested revisions (lines 37–38).

Comment 3: Lines 45-46: According to this classification system, 35 G- and 50 P- genotypes have been described globally. Currently, 42 G- and 58 P-genotypes are described. https://rega.kuleuven.be/cev/viralmetagenomics/virus-classification/rcwg.

Answer: Thank you, we have made the suggested revision (lines 45–46).

 Results:

Comment 4: Page 3, lines 130-137: The authors comment on positivity by geographic region over 5 years and annual positivity rates. However, these data are not clearly shown in the tables. The tables are confusing. It is unclear how many samples were obtained by region/year.

Answer: We have revised Table 2 and added a new Figure 1.

Comment 5: Perhaps it would be better if Table 1 presented the overall results showing the number of tested/positive samples by year and region.

Answer: Thank you. As mentioned above, we have revised Table 2 and add a new Figure 1.

Comment 6: In Table 2, the genotyping result could be introduced. And maybe a graph showing the regions and genotypes identified by region/year.

Answer: In accordance with the reviewer’s comment, we have revised Table 2 and added a new Figure 1.

Comment 7: Page 6, lines 163-169: Please rephrase. Based on the data presented, although genotypes G3, G5, G11 and G26 were detected, they were not "most prevalent". The G4 and G9 genotypes were the most prevalent. The same occurred with respect to genotypes P[13], P[23] and P[26].

Answer: Thank you. We have revised the text accordingly (lines 158–164, 271-272).

Discussion:

Comment 8: The discussion needs to be greatly improved. The text is repetitive. It is not necessary to present the detection percentages for each genotype in different countries. The authors have already presented, in the introduction, which RVA genotypes are most prevalent in the world. It would be more fruitful to focus the discussion on the evolution of epidemiology in South Korea and the implications of this in relation to the development of measures to prevent and control these infections.

Answer: Thank you. We have revised the Discussion section as requested.

Round 2

Reviewer 2 Report

The major concerns of the reviewer have been addressed except the following minor concern.

 Minor concern:

 Line 159, “Among them, G4 and G9 were the prevalent VP7 genotypes should be “Among them, G4 and G9 were the predominant VP7 genotypes”.

Author Response

Reviewer 2

Minor concern:

Comment 1: Line 159, “Among them, G4 and G9 were the prevalent VP7 genotypes” should be “Among them, G4 and G9 were the predominant VP7 genotypes”.

Answer: We have changed the sentence to “Among them, G4 and G9 were the predominant VP7 genotypes” (line 151).

Additional comments are described as below.

Comment 1: P1, L39: The author should split up the paragraph at this point.

Answer: We have divided the paragraph (line 38).

Comment 2: Please include subtotal and proportion in table 1.

Answer: We have included the subtotal and proportion in Table 1.

Comment 3: In table 2, the author should present the number of positive/tested

samples by year and region.

Answer: We have presented the number of positive/tested samples by year and region in Table 2.

Comment 4: P9, L244-249: The author should summarize the result of previous study.

Answer: We have revised the sentences (lines 242–246).

Comment 5: P9, L253-257: The author should summarize the result of previous study. In

addition, please check the reference you mentioned.

Answer: We have revised the sentences (lines 249–256).

Comment 6: P9, L257-260: In India, identification of novel genotype G26 was already

reported. It would be better the author compare the sequence of G26 isolates

in this study with that of isolates in Japan and India. And then the author

should discuss the meaning of the detection of G26 in South Korea.

Answer: We have revised the sentences (lines 249–256).

Comment 7: P9, L260: The author should split up the paragraph at this point.

Answer: We have divided the paragraph (line 257).

Comment 8: P9, L260-267: The author should avoid duplicate repeated text. I was

surprised to see the zoonotic potential of the G4 genotype was not touched on

the discussion.

Answer: We have revised the sentences (lines 257–264).

Comment 9: P9, L265-267: Please remove the sentence. The author should focus on

discussing the experimental results obtained in this study.

Answer: We have revised the sentences (lines 261-264).

Comment 10: P9, L268-270: This part seems the description of the literature unrelated

to the experimental results.

Answer: We have removed the sentence.

Comment 11: P9, L270-273: The author should summarize the result of previous study.

Answer: We have revised the sentence (lines 265–267).

Comment 12: P10, L279-282: Please remove the sentence about unrelated literature on

this part. Moreover, the author should provide the basis of expectation.

Answer: We have removed the sentences.

Comment 13: P10, L282-288: I found similar text in reference 14. The author should

focus on discussing the experimental results obtained in this study.

Answer: We have revised the sentences (lines 274–279).

Comment 14: P10, L289-300: The author should summarize the result of previous study

and provide the rationale for the need to develop new inactivated vaccine.

Answer: We have revised the sentences (lines 280–286).

Comment 15: Please remove 'Isolation and' from the title.

Answer: We have removed 'Isolation and' from the title.
